# Are socioeconomic inequalities in the incidence of small-for-gestational-age birth narrowing? Findings from a population-based cohort in the South of England

Sam Wilding,[1] Nida Ziauddeen,[1] Paul Roderick,[1] Dianna Smith,[2] Debbie Chase,[3] Nick Macklon,[4,5] Nuala McGrath,[1,6] Mark Hanson,[7,8] Nisreen A Alwan[1,8]

For numbered affiliations see end of article.

**Correspondence to**
Dr Nisreen A Alwan;
N.A.Alwan@soton.ac.uk

## ABSTRACT

**Objectives** To investigate socioeconomic inequalities, using maternal educational attainment, maternal and partner employment status, and lone motherhood indicators, in the risk of small-for-gestational-age (SGA) births, their time trend, potential mediation by maternal smoking and body mass index, and effect modification by parity.

**Design** Population-based birth cohort using routine antenatal healthcare data.

**Setting** Babies born at University Hospital Southampton, UK, between 2004 and 2016.

**Participants** 65 909 singleton live births born to mothers aged ≥18 years between 24-week and 42-week gestation.

**Main outcome measures** SGA (birth weight <10th percentile for others born at the same number of completed weeks compared with 2013/2014 within England and Wales).

**Results** Babies born to mothers educated up to secondary school level (adjusted OR (aOR) 1.32, 99% CI 1.19 to 1.47), who were unemployed (aOR 1.27, 99% CI 1.16 to 1.38) or with unemployed partners (aOR 1.27, 99% CI 1.13 to 1.43), were at greater risk of being SGA. There was no statistically significant change in the magnitude of this risk difference by these indicators over time between 2004 and 2016, as estimated by linear interactions with year of birth. Babies born to lone mothers were not at higher risk compared with partnered mothers after adjusting for maternal smoking (aOR 1.05, 99% CI 0.93 to 1.20). The inverse association between maternal educational attainment and SGA risk appeared greater in multiparous (aOR 1.40, 99% CI 1.10 to 1.77) compared with primiparous women (aOR 1.28, 99% CI 1.12 to 1.47), and the reverse was true for maternal and partner's unemployment where the association was stronger in primiparous women.

**Conclusions** Socioeconomic inequalities in SGA risk by educational attainment and employment status are not narrowing over time, with differences in association strength by parity. The greater SGA risk in lone mothers was potentially explained by maternal smoking. Preventive interventions should target socially disadvantaged women, including preconception and postpartum smoking cessation to reduce SGA risk.

## Strengths and limitations of this study

► This study uses a relatively large sample of population-level antenatal care data to examine the risk of small-for-gestational-age births by socioeconomic factors.

► Standard routinely collected measures recorded at the first antenatal appointment are utilised which can be used for risk prediction in practice without the need to collect extra data during antenatal appointments.

► Limitations include the transferability of results from this population to others with differing characteristics, that socioeconomic factors were only assessed at one time point in pregnancy, and self-reporting of educational qualifications and employment.

## INTRODUCTION

Babies born small for gestational age (SGA) are at higher risk of neonatal morbidity, mortality[1] and childhood obesity potentially through compensatory early growth.[2 3] Numerous clinical and lifestyle risk factors are associated with the risk of being SGA, including maternal height, weight, diet, ethnicity, parity, smoking, pre-eclampsia and hypertension.[4 5] Closely linked to these risk factors there is extensive evidence of socioeconomic status (SES) inequalities, with more SGA babies born to mothers living in the most deprived communities compared with those in the most affluent.[6]

Several proxies of SES are present in the literature, with area measures of wealth, maternal education, employment and income being the most common indicators, while paternal factors being notably absent.[7] Disadvantaged SES groups (in terms of education and income) typically experience greater rates of SGA births.[8 9] The majority of

studies rely on one proxy of SES, but studies controlling for several SES measures find that different aspects of SES are independently associated with the risk of SGA.[10–14]

Despite the wealth of research on the association between parental SES and SGA, the underlying mechanisms are poorly understood.[15] Current explanations focus on the availability of (physiological and material) resources and mediating factors that differ between women of high and low SES. For resources, the 'weathering' hypothesis states that women in low SES at the time of conception have experienced relatively high levels of cumulative disadvantage in terms of income, stress and diet, which have led to a deterioration in physiological health.[16] This association may also be mediated by lifestyle factors, wherein mothers in low SES are more likely to be exposed to or partake in risk factors for SGA such as smoking. Mediation analyses have found that higher rates of underweight and smoking at conception among mothers with low educational attainment mediate the association between SES and birth outcomes in the UK.[15 17]

The extent of these SES inequalities in the risk of SGA may differ between first and higher order births. The birth of the first child brings significant physiological, wellbeing and social changes,[18] and women in low SES may have weaker social support mechanisms to adjust to these changes, as they appear to be at higher risk of SGA in subsequent births after adjusting for clinical risk factors.[19] Risk factors for SGA specific to second and higher order births are more prevalent in women of low SES, with postnatal depression being more common in mothers without a university degree and those in poverty.[20 21]

In England, public health policy aims to narrow SES inequalities in birth outcomes over time,[22 23] and changes in the extent of inequalities in SGA have been noted in other European countries since the early 2000s.[24] Major welfare reforms enacted in the UK between 1999 and 2002 increased in-work tax incentives, which particularly increased the net income of part-time working women, relative to those out of work.[25] In 2008, the global 'great recession' occurred, after which single mothers in England became increasingly less likely to be employed, while facing disproportionate losses of welfare income, facing a double income penalty relative to working mothers.[26] The recession appears to have had differential impacts on women by level of educational attainment, with those without a university degree experiencing a postrecession rise in the prevalence of obesity, relative to those with degrees.[27]

Utilising an antenatal healthcare database in Hampshire, England, we aimed to examine differences in SGA risk by SES indicators, investigate if these differences are mediated by maternal body mass index (BMI) and smoking, and whether the inequalities gap has narrowed over the 13 year study period (2004–2016). In addition, we aimed to stratify by parity in order to examine whether the SES gap in SGA risk is the same at first births, relative to second and higher order births.

## METHODS
### Data
This analysis is based on a population-based cohort including anonymised antenatal and birth records of women aged ≥18 years who had a live singleton birth between 1 January 2004 and 31 December 2016 at the University Hospital Southampton (UHS) National Health Service (NHS) Trust in the South of England. UHS is the primary centre for maternity care for the city of Southampton and the surrounding areas, and is the regional centre for high-risk pregnancies. The process of deriving a sample for analysis is outlined in online supplementary figure 1. To ensure that the findings are applicable to the majority of (non-high-risk) pregnancies, records with late first antenatal (booking) appointments (after 24-week gestation, as assessed by ultrasound) and of mothers under the age of 18 were excluded. First, we analysed the risk of SGA by SES in all births (including more than one birth per mother if in the database and study timeframe), adjusting for confounding and clustering. We then tested whether differences between SES groups (by maternal education, employment, paternal employment and partnership status) have changed over the study period (2004–2016). We then limited the analysis to the first recorded birth per mother in the dataset, and stratified by parity (primiparous and multiparous), to avoid biasing subanalyses via double counting.

### Assessment of SES exposures
Socioeconomic measures were self-reported at the first antenatal (booking) appointment, which is recommended by the National Institute for Health and Care Excellence Antenatal Care Guidelines to occur by the 10th week of gestation.[28] Mothers were asked to report their highest educational qualification, classified as university degree (highest level), college (A levels) or secondary school (General Certificate of Secondary Education), whether they were currently employed, and if their partners were currently employed (possible answers included employed, unemployed and seeking work or student, with the latter two being combined). Partnership status was self-reported at the same appointment. All four SES proxies were categorised to be compared with mothers with advantaged SES (mother has a university degree; mother is employed; mother's partner is employed; mother has a partner). Time trends in SES factors were examined, and presented in online supplementary figure 2.

### Assessment of outcome
Birth weight was measured by healthcare professionals for all births in the dataset. Gestational age was based on a dating ultrasound scan performed by healthcare professionals, and was present for all records in the dataset. Birth weight centile for gestational age is calculated using reference values provided in the most recently released data (2013–2014) for England and Wales, which were validated using 2015 records.[29] Given that the association

between SES and preterm births is well established in the literature,[30] and that gestational age is strongly associated with birth weight, we used a SGA measure to assess low birth weight rather than the standard birth weight cut-off.

The birth centile references are available for 24–42 completed weeks of gestation, so live births at ≤23 (71) or >42 (568) completed weeks or with indeterminate sex (16) are excluded from the analysis (SGA sample=65 909/66 564). In line with WHO guidelines, UK guidelines and common practice, SGA is defined as a birth weight lower than the 10th percentile compared with others born at the same number of weeks gestation in the sex-specific reference centiles,[31–33] and all others are defined as not small for gestational age (non-SGA).

### Assessment of confounder and mediator variables

Maternal age, height, parity, ethnicity and smoking history were self-reported at the booking antenatal appointment. Baby's sex was recorded at birth by a healthcare professional. Maternal weight and blood pressure were measured by a healthcare professional at the booking appointment, and screening for gestational diabetes was carried out for women identified as at high risk in the second trimester of pregnancy.[28] Maternal age, ethnicity, gestational diabetes, gestational hypertension and systolic blood pressure at booking were adjusted for in the multivariable models, as these factors have been associated with size at birth in previous analyses.[4 34 35] Parity (no versus one or more previous births) was treated as a confounder in the models analysing the whole sample, and then as an effect modifier for SES through interaction terms and later stratification. Maternal BMI and smoking history are included as potential mediators of the relationship between SES and risk of SGA, based on previous evidence.[15 17] Maternal BMI was categorised as underweight (<18.5), normal (18.5–24.9), overweight/preobese (25.0–29.9) and obese (30+),[36] and treated as a categorical variable in all analyses. Maternal smoking was reported as follows (never smoked, ex-smoker, <10 per day, 10–20 per day and 20+ per day), and also treated as a categorical variable in all analyses.

### Patient and public involvement

This analysis uses routinely collected antenatal data where patient identifiers were anonymised. No patients or members of the public were recruited or consulted by the research team.

### Statistical analysis

All analyses were conducted using Stata V.15. Descriptive statistics and the unadjusted ORs between all variables and risk of SGA are presented in table 1. T-tests were used to test whether the mean of each continuous variable (maternal BMI, age and systolic blood pressure at booking) differed between those born SGA and non-SGA. Multivariable logistic regression models were used to estimate ORs, p values and respective 99% CI for SES differences in the risk of SGA independently

after adjustment for control variables, after adjustment for other SES indicators, and then after controlling for mediators. A p value cut-off of 0.01 is used to test for statistical significance when reporting risk rather than the more conventional 0.05 cut-off in order to minimise the risk of type I error due to multiple testing, as adjusted models control for multiple SES indicators.[37] Evidence of mediation is examined through assessing the attenuation of SES with SGA associations once known risk factors are controlled for, and the significance once each a priori mediator (first BMI, then smoking) is controlled for.[38] In all logistic regressions, cases with missing data for variables within the model were dropped (complete case analysis).

In the first analysis, adjusted ORs (aORs) for the risk of a baby being born SGA are presented in model 1 (control variables include maternal age, parity, ethnicity, gestational diabetes, gestational hypertension and systolic blood pressure at booking) independently for maternal education, employment and partnership status, adjusting for clustering of births within the same mother. In model 2, all three of these SES proxies are controlled for, in addition to the control variables in model 1, before including the two mediators (maternal BMI and smoking) sequentially in models 3 and 4. Due to collinearity between maternal partnership status and partner's employment, the association for the latter is tested separately with the same structure.

In the second analysis, year and the interactions between year and SES indicator (slope) effects are included to model 4 for maternal education, employment, partner's employment and partnership status, to test whether SES inequalities in the risk of being born SGA are widening or narrowing over time during the study period. These slopes represent the change in relative odds of SGA for the socioeconomic group relative to the control group for each year in the dataset (2004–2016). ORs>1 indicate that this group became at higher risk of SGA births over time, relative to the control group.[39] Further models were estimated including SES interactions between a dummy indicator for records pre-2008 (2004–2008) and post-2008 (2009–2016), to test whether SES inequalities in the risk of SGA changed in magnitude between the two periods.

In the third analysis, the sample is limited to the first birth for each mother (one birth per mother), and then stratified by parity (primiparous or multiparous). Limiting the sample to the first birth for each mother acts as a sensitivity analysis for the first analysis, ensuring that the results are not influenced by multiple births per mother. Interactions between SES and parity are estimated to test whether the association between SES and risk of SGA is modified by parity, and then parity-stratified modelling was conducted. A p value cut-off of 0.05 is used to test for interactions. As in the first analysis, adjusted SES ORs are presented for each subsample, then these ORs are adjusted for other SES indicators, before including mediators (maternal BMI and smoking).

**Table 1** Maternal/pregnancy characteristics by small-for-gestational-age (SGA) status (birth weight <10th percentile for gestational age) in the University Hospital Southampton (UHS) maternity population-based cohort (singleton live births 2004–2016, n=65 909)

| Characteristics | SGA n | % | Non-SGA n | % | % SGA % SGA | 99% CI |
|---|---|---|---|---|---|---|
| **Highest qualification** | | | | | | |
| University degree or higher | 1550 | 24.4 | 17 518 | 29.5 | 8.1 | 7.6 to 8.6 |
| College | 2378 | 37.4 | 24 011 | 40.4 | 9.0 | 8.5 to 9.5 |
| Secondary school or lower | 2429 | 38.2 | 17 958 | 30.2 | 11.9 | 11.3 to 12.5 |
| **Maternal employment** | | | | | | |
| Employed | 3877 | 61.3 | 40 561 | 68.6 | 8.7 | 8.4 to 9.1 |
| Unemployed | 2446 | 38.7 | 18 533 | 31.4 | 11.6 | 11.1 to 12.2 |
| **Partner's employment** | | | | | | |
| Employed | 4981 | 85.6 | 50 675 | 90.9 | 8.9 | 8.6 to 9.3 |
| Unemployed | 841 | 14.4 | 5079 | 9.1 | 14.2 | 13.0 to 15.4 |
| **Partnership** | | | | | | |
| Partnered | 5721 | 90.0 | 55 054 | 92.5 | 9.4 | 9.1 to 9.7 |
| Lone mother | 639 | 10.0 | 4495 | 7.6 | 12.4 | 11.3 to 13.7 |
| **Maternal BMI** | | | | | | |
| <18.5 (underweight) | 395 | 6.2 | 1586 | 2.7 | 19.9 | 17.6 to 22.3 |
| 18.5–24.9 (normal weight) | 3641 | 57.3 | 30 758 | 51.7 | 10.6 | 10.1 to 11 |
| 25–29.9 (overweight) | 1425 | 22.4 | 16 083 | 27.0 | 8.1 | 7.6 to 8.7 |
| 30+ (obese) | 899 | 14.1 | 11 122 | 18.7 | 7.5 | 6.9 to 8.1 |
| **Smoking** | | | | | | |
| Never smoked | 3059 | 48.1 | 30 791 | 51.8 | 9.0 | 8.6 to 9.4 |
| Ex-smoker | 1492 | 23.5 | 19 758 | 33.2 | 7.0 | 6.6 to 7.5 |
| <10 per day | 1040 | 16.4 | 5557 | 9.4 | 15.8 | 14.6 to 17.0 |
| 10–20 per day | 694 | 10.9 | 3103 | 5.2 | 18.2 | 16.6 to 19.9 |
| >20 per day | 71 | 1.1 | 252 | 0.4 | 22.0 | 16.3 to 28.5 |
| **Maternal age** | | | | | | |
| 18–24 | 2004 | 31.5 | 14 364 | 24.1 | 12.2 | 11.6 to 12.9 |
| 25–34 | 3428 | 53.9 | 35 678 | 59.9 | 8.8 | 8.4 to 9.1 |
| 35–39 | 758 | 11.9 | 8004 | 13.5 | 8.6 | 7.9 to 9.4 |
| 40+ | 170 | 2.7 | 1503 | 2.5 | 10.1 | 8.3 to 12.2 |
| **Previous live births** | | | | | | |
| None | 3526 | 55.4 | 25 136 | 42.2 | 12.3 | 11.8 to 12.8 |
| One or more | 2834 | 44.6 | 34 413 | 57.8 | 7.6 | 7.2 to 8 |
| **Maternal ethnicity** | | | | | | |
| White | 4807 | 75.6 | 49 531 | 83.2 | 8.8 | 8.5 to 9.2 |
| Mixed | 87 | 1.4 | 721 | 1.2 | 10.8 | 8.1 to 13.9 |
| Asian | 810 | 12.8 | 3622 | 6.1 | 18.3 | 16.8 to 19.8 |
| Black/African/Caribbean | 148 | 2.3 | 1096 | 1.8 | 11.9 | 9.6 to 14.4 |
| Chinese | 31 | 0.5 | 429 | 0.7 | 6.8 | 4.1 to 10.4 |
| Other | 116 | 1.8 | 832 | 1.4 | 12.2 | 9.6 to 15.2 |
| Not known | 361 | 5.7 | 3318 | 5.6 | 9.8 | 8.6 to 11.1 |
| **Gestational diabetes** | | | | | | |
| Not present in current pregnancy | 6213 | 97.7 | 58 074 | 97.5 | 9.7 | 9.4 to 10.0 |

**Table 1** Continued

| Characteristics | SGA | | Non-SGA | | % SGA | |
|---|---|---|---|---|---|---|
| | n | % | n | % | % SGA | 99% CI |
| Present in current pregnancy | 147 | 2.3 | 1475 | 2.5 | 9.0 | 7.3 to 11 |
| Systolic blood pressure | | | | | | |
| <140 mm Hg | 6219 | 99.0 | 57 831 | 98.6 | 9.7 | 9.4 to 10 |
| >=140 mm Hg | 64 | 1.0 | 812 | 1.4 | 7.3 | 5.2 to 9.9 |
| Overall | 6360 | 100 | 59 549 | 100 | 9.7 | 9.4 to 9.9 |

| | SGA | | Non-SGA | | |
|---|---|---|---|---|---|
| | Mean | SD | Mean | SD | P value for t-test |
| Maternal BMI | 24.5 | 5.3 | 25.7 | 5.5 | <0.001 |
| Maternal age | 27.9 | 5.8 | 28.8 | 5.5 | <0.001 |
| Maternal systolic blood pressure | 108.7 | 11.5 | 109.9 | 11.3 | <0.001 |

Source: UHS antenatal records for live singleton births (2004–2016). Records with a late antenatal booking (over 24-week gestation) were excluded. Variables with missing information include maternal education (65), maternal employment (492), maternal smoking (92), maternal systolic blood pressure (983) and partner's employment (4333). The percentage SGA column indicates the percentage of babies born SGA for this characteristic, and the accompanying 99% CI. The t-test indicates whether the mean of each variable differs between those born SGA and non-SGA.
BMI, body mass index.

## RESULTS

There are 65 909 singleton live births within the dataset which can be categorised as SGA or non-SGA to 44 428 mothers. Of births, 71% were to women with no university degree, in employment (67.9%), have partners the time of booking (92.3%), who are in employment (90.4%), of white ethnicity (82.4%) and with normal (<140 mm Hg) systolic blood pressure (98.7%). Of these 65 909 births, 6360 (9.7%, 99% CI 9.4 to 9.9%) were born SGA (table 1).

Time trends in SES factors are displayed in online supplementary figure 1. Briefly, less than college (A levels) educational qualification, maternal unemployment and lone motherhood became less prevalent over time (39%, 34% and 9% in 2004 to 22%, 29% and 6% in 2016, respectively), while partner unemployment remained relatively stable.

The proportion of SGA births was higher than the average for births to mothers in all disadvantaged SES groups. This includes births to mothers with no university degree (college qualification: 9.0% born SGA, 99% CI 8.5 to 9.5, secondary school qualification: 11.9% born SGA, 99% CI 11.3 to 12.5), births to unemployed mothers (11.6% born SGA, 99% CI 11.1 to 12.2), births to mothers with unemployed partners (14.2% born SGA, 99% CI 13.0 to 15.4) and births to single mothers (12.4% born SGA, 99% CI 11.3 to 13.7). Other maternal factors associated with a higher than average rate of SGA include maternal BMI <18.5 kg/m$^2$ (19.9% born SGA, 99% CI 17.6 to 22.3), maternal smoking at booking (16.8% born SGA, 99% CI 15.9 to 17.8) and Asian ethnicity (18.3% born SGA, 99% CI 16.8 to 19.8).

### SES differences in SGA risk in the whole cohort

Estimates of the association between maternal SES indicators and risk of SGA are presented in table 2. The univariable associations between each SES indicator and the risk of SGA are presented in the unadjusted risk row, with all SES indicators being associated with SGA. The size of these effects increases in the first adjusted model (controlling for maternal age, ethnicity, parity, gestational diabetes, gestational hypertension and systolic blood pressure at booking), and attenuate once other SES indicators are controlled for (model 2). Accounting for maternal BMI class had limited impact on effect sizes (model 3). After including maternal smoking, all SES inequalities reduced in size (model 4), with the ORs for college qualification compared with university degree (OR 1.11, 99% CI 1.00 to 1.22) and lone motherhood compared with partnered status attenuating at the 99% level (OR 1.05, 99% CI 0.93 to 1.20). The full results for model 4 are presented in online supplementary table 1.

In unadjusted estimates presented in table 3, those born to mothers with unemployed partners at the antenatal booking appointment are 68% more likely to be born SGA (OR 1.68 99% CI 1.51 to 1.88) in comparison to those born to mothers with employed partners. This association slightly attenuates once maternal education and employment are controlled for (model 2). The association attenuates further once maternal BMI is controlled for (model 3) and remains similar once smoking is accounted for (model 4 OR 1.27, 99% CI 1.13 to 1.43). The full results for model 4 are presented in online supplementary table 2.

As a sensitivity analysis, we repeated the modelling for a subgroup of women who were resident in Southampton

**Table 2** Risk of being born small for gestational age (birth weight <10th percentile for gestational age) by maternal socioeconomic indicator in the University Hospital Southampton maternity population-based cohort (singleton live births 2004–2016)

| | Mothers with a college qualification versus university degree | | | Mothers with a school qualification versus university degree | | | Mothers unemployed at the first antenatal appointment | | | Lone mothers at the first antenatal appointment | | |
|---|---|---|---|---|---|---|---|---|---|---|---|---|
| | OR | 99% CI | P value | OR | 99% CI | P value | OR | 99% CI | P value | OR | 99% CI | P value |
| Unadjusted risk | 1.12 | 1.02 to 1.23 | 0.002 | 1.53 | 1.39 to 1.68 | <0.001 | 1.38 | 1.28 to 1.49 | <0.001 | 1.37 | 1.22 to 1.54 | <0.001 |
| Adjusted risk—model 1 | 1.20 | 1.09 to 1.32 | <0.001 | 1.65 | 1.50 to 1.83 | <0.001 | 1.56 | 1.43 to 1.69 | <0.001 | 1.41 | 1.25 to 1.59 | <0.001 |
| Adjusted risk—model 2 | 1.17 | 1.06 to 1.29 | <0.001 | 1.52 | 1.38 to 1.68 | <0.001 | 1.43 | 1.31 to 1.56 | <0.001 | 1.25 | 1.11 to 1.41 | <0.001 |
| Adjusted risk—model 3 | 1.20 | 1.09 to 1.32 | <0.001 | 1.56 | 1.41 to 1.73 | <0.001 | 1.42 | 1.30 to 1.55 | <0.001 | 1.24 | 1.10 to 1.41 | <0.001 |
| Adjusted risk—model 4 | 1.11 | 1.00 to 1.22 | 0.009 | 1.32 | 1.19 to 1.47 | <0.001 | 1.27 | 1.16 to 1.38 | <0.001 | 1.05 | 0.93 to 1.20 | 0.280 |

Model 1 adjusts for maternal age, ethnicity, parity, gestational diabetes and systolic blood pressure.
Model 2 is model 1 plus the other socioeconomic status indicators (n births=64 535, n mothers=43 787).
Model 3 is model 2 plus maternal body mass index as a potential mediator (n births=64 535, n mothers=43 787).
Model 4 is model 3 plus maternal smoking history as an additional mediator (n births=64 535, n mothers=43 787).
In all models, the SEs are adjusted for multiple births per mother.

**Table 3** Risk of being born small for gestational age (birth weight <10th percentile for gestational age) by partner's employment status in the University Hospital Southampton maternity population-based cohort (singleton live births 2004–2016)

| | Mothers with an unemployed partner | | |
|---|---|---|---|
| | OR | 99% CI | P value |
| Unadjusted risk | 1.68 | 1.51 to 1.88 | <0.001 |
| Adjusted risk—model 1 | 1.69 | 1.51 to 1.90 | <0.001 |
| Adjusted risk—model 2 | 1.48 | 1.31 to 1.66 | <0.001 |
| Adjusted risk—model 3 | 1.49 | 1.33 to 1.67 | <0.001 |
| Adjusted risk—model 4 | 1.27 | 1.13 to 1.43 | <0.001 |

Model 1 adjusts for maternal age, ethnicity, parity, gestational diabetes and systolic blood pressure.
Model 2 is model 1 plus the other two socioeconomic status indicators (n births=60 385, n mothers=41 841).
Model 3 is model 2 plus maternal body mass index as a potential mediator (n births=60 385, n mothers=41 841).
Model 4 is model 3 plus maternal smoking history as an additional mediator (n births=60 385, n mothers=41 841).
In all models, the SEs are adjusted for multiple births per mother.

at the time of delivery to address the potential that the whole sample results may be biased by including potential high-risk referrals from other regions to this specialised maternity centre. The geographical residence data (lower super output areas) were retrieved from health visitor records, and linked to births in this cohort as part of a bigger research project using an anonymised linked mother–child dataset. Each child in England and Wales is followed up by health visitor teams for at least five key appointments which start at 28 weeks into pregnancy,[40] so this subsample is unlikely to be affected by selection bias. From the sample of 64 535 births, 30 663 (47.5%) were resident in Southampton at this 28-week appointment. In a model that adjusts for all confounders, maternal BMI category and smoking, the CIs for the SES factors overlap in the Southampton-only sample, and those results presented in tables 2 and 3 (see online supplementary table 3 for full results), indicating largely similar risk estimates between the two samples.

### Time trend in SES inequalities in the risk of SGA between 2004 and 2016

To test whether SES inequalities are narrowing or widening over time, interactions between year (continuous) and SES ('slope') were included to model 4 in tables 2 and 3, and expressed as ORs. A positive slope OR indicates that the disadvantaged SES group is becoming at greater risk of SGA relative to the advantaged group over calendar year, and vice versa for a negative effect.

Figure 1A–D displays the adjusted ORs for each SES indicator by year in the cohort (UHS), and the accompanying p value for the slope over calendar year. The slopes for maternal college and school qualifications (OR 1.00, 99% CI 0.97 to 1.02; OR 1.00, 99% CI 0.97 to

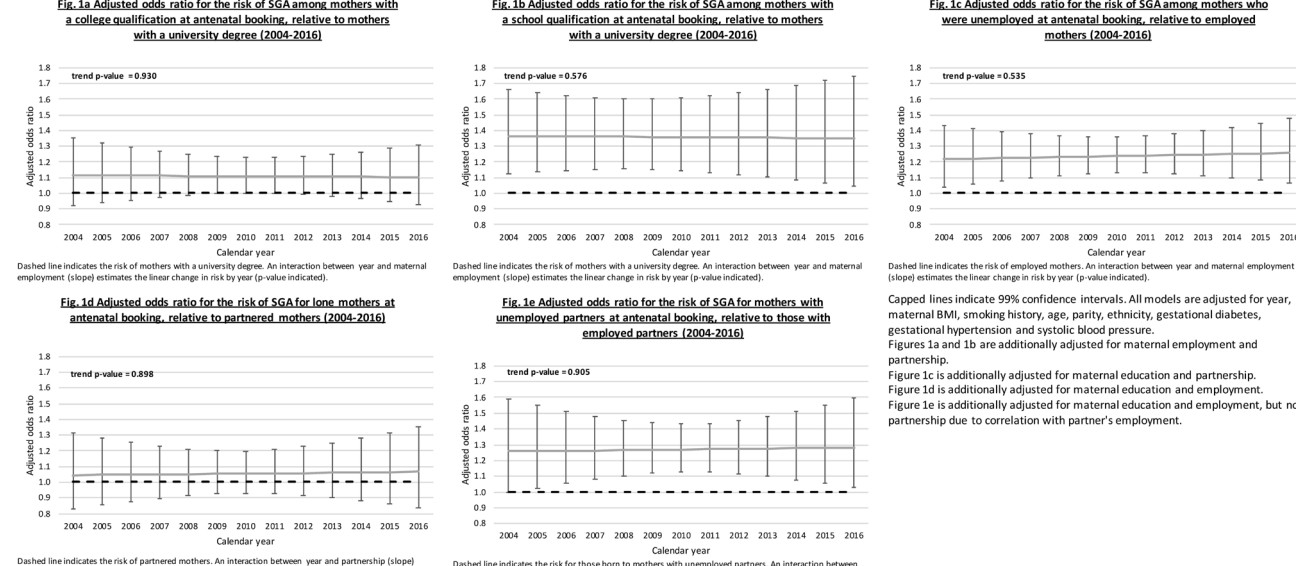

**Figure 1** Risk of being born small for gestational age (SGA) (birth weight <10th percentile for gestational age) by parental socioeconomic status indicators in the University Hospital Southampton (UHS) maternity population-based cohort (singleton live births 2004–2016).

1.02), maternal employment (OR 1.01, 99% CI 0.98 to 1.03), lone motherhood (OR 1.00, 99% CI 0.97 to 1.03) and partner unemployment (OR 1.00, 99% CI 0.97 to 1.03) were not statistically significant. Models using a binary indicator for pre-2008 and post-2008 (2003–2008 and 2009–2016) showed no significant differences in the magnitude of SES inequalities (results not shown).

### SES differences in SGA risk by maternal parity status

For this analysis, the sample was restricted to the first antenatal care record per mother included in our dataset with no missing information (20 748 records dropped, with a new total of 43 787). Interaction terms between each SES indicator and parity (accounting for control variables) were conducted using this sample showing a significant interaction between maternal employment status and SGA (p=0.010). We then stratified the sample by parity (n primiparous (0 previous live births)=28 171; n multiparous (1 or more previous live births)=15 616). The modelling strategy used in the first analysis is repeated on these subsamples to assess the risk estimates by parity.

The association between secondary school qualification versus university degree and the risk of SGA appeared less pronounced among primiparous (OR 1.28, 99% CI 1.12 to 1.47) than multiparous women (OR 1.40, 99% CI 1.10 to 1.77). Maternal unemployment (relative to mothers who were employed) was associated with higher risk of SGA in primiparous women (aOR 1.29, 99% CI 1.13 to 1.46) than among multiparous women (aOR 1.17, 99% CI 0.99 to 1.38). The associations between college qualification versus university degree, and lone motherhood versus partnered status, with SGA risk appeared to be mediated by smoking in all subsamples (table 4).

Table 5 displays the results for partner's employment (total n mothers=41 841; 26 498 primiparous,

15 343 multiparous). The association between partner's employment and risk of SGA appeared to be mediated by maternal smoking among multiparous women (aOR 1.15, 99% CI 0.93 to 1.43), but not primiparous women (aOR 1.33, 99% CI 1.12 to 1.58). The estimates of SES differences in the risk of SGA were similar in the reduced sample (tables 4 and 5) and the whole sample (tables 2 and 3).

To summarise the above models, both maternal and partner's employment status appeared to be more strongly associated with SGA risk in primiparous than multiparous women, and the reverse is true for maternal educational attainment.

### DISCUSSION

In this analysis of routine maternity healthcare data from a regional hospital in Southampton, UK, multivariable logistic regression was used to examine the relationship between SES indicators (education, employment and partnership) and SGA, and whether these relationships are stable over time and different by parity. Educational attainment and employment (of the mother and her partner) were independently associated with the risk of SGA, although differences between the association between single motherhood and SGA were attenuated by adjusting for smoking status. SES differences in the risk of SGA were stable over the study period (2004–2016). The strength of these SES differences varied between mothers at their first and higher order births. Maternal and partner unemployment were associated with a higher risk of SGA in mothers with no previous live births, with lower educational qualification being more strongly associated with SGA risk in mothers with previous live births.

**Table 4** Risk of being born small for gestational age (birth weight <10th percentile for gestational age) by maternal socioeconomic indicator and stratified by parity in the University Hospital Southampton maternity population-based cohort (singleton live births 2004–2016, one birth per mother)

| Sample | Model | Mothers with a college qualification versus university degree | | | Mothers with a school qualification versus university degree | | | Mothers unemployed at the first antenatal appointment | | | Lone mothers at the first antenatal appointment | | |
|---|---|---|---|---|---|---|---|---|---|---|---|---|---|
| | | OR | 99% CI | P value | OR | 99% CI | P value | OR | 99% CI | P value | OR | 99% CI | P value |
| Whole sample, n mothers=43 787 | Unadjusted risk | 1.09 | 0.98 to 1.20 | 0.029 | 1.43 | 1.29 to 1.58 | <0.001 | 1.37 | 1.26 to 1.49 | <0.001 | 1.33 | 1.16 to 1.52 | <0.001 |
| | Adjusted risk—model 1 | 1.18 | 1.06 to 1.31 | <0.001 | 1.58 | 1.41 to 1.76 | <0.001 | 1.48 | 1.35 to 1.64 | <0.001 | 1.40 | 1.22 to 1.60 | <0.001 |
| | Adjusted risk—model 2 | 1.17 | 1.054 to 1.30 | <0.001 | 1.49 | 1.33 to 1.66 | <0.001 | 1.38 | 1.26 to 1.53 | <0.001 | 1.26 | 1.10 to 1.45 | <0.001 |
| | Adjusted risk—model 3 | 1.19 | 1.07 to 1.33 | <0.001 | 1.51 | 1.35 to 1.69 | <0.001 | 1.37 | 1.24 to 1.51 | <0.001 | 1.26 | 1.10 to 1.45 | <0.001 |
| | Adjusted risk—model 4 | 1.11 | 1.00 to 1.24 | 0.012 | 1.31 | 1.17 to 1.47 | <0.001 | 1.24 | 1.12 to 1.37 | <0.001 | 1.09 | 0.95 to 1.25 | 0.122 |
| Primiparous women only, n births=28 171 | Unadjusted risk | 1.11 | 0.99 to 1.25 | 0.015 | 1.50 | 1.33 to 1.69 | <0.001 | 1.78 | 1.60 to 1.99 | <0.001 | 1.32 | 1.11 to 1.55 | <0.001 |
| | Adjusted risk—model 1 | 1.14 | 1.01 to 1.29 | 0.005 | 1.47 | 1.29 to 1.67 | <0.001 | 1.50 | 1.32 to 1.69 | <0.001 | 1.32 | 1.11 to 1.56 | <0.001 |
| | Adjusted risk—model 2 | 1.13 | 1.00 to 1.28 | 0.008 | 1.40 | 1.23 to 1.60 | <0.001 | 1.42 | 1.25 to 1.61 | <0.001 | 1.20 | 1.01 to 1.43 | 0.006 |
| | Adjusted risk—model 3 | 1.15 | 1.02 to 1.30 | 0.002 | 1.41 | 1.24 to 1.61 | <0.001 | 1.39 | 1.23 to 1.58 | <0.001 | 1.20 | 1.01 to 1.43 | 0.006 |
| | Adjusted risk—model 4 | 1.09 | 0.97 to 1.24 | 0.057 | 1.28 | 1.12 to 1.47 | <0.001 | 1.29 | 1.13 to 1.46 | <0.001 | 1.07 | 0.90 to 1.28 | 0.289 |
| Multiparous women only, n births=15 616 | Unadjusted risk | 1.24 | 0.99 to 1.55 | 0.014 | 1.82 | 1.47 to 2.25 | <0.001 | 1.52 | 1.31 to 1.77 | <0.001 | 1.49 | 1.20 to 1.86 | <0.001 |
| | Adjusted risk—model 1 | 1.35 | 1.07 to 1.70 | 0.001 | 1.92 | 1.53 to 2.40 | <0.001 | 1.45 | 1.23 to 1.69 | <0.001 | 1.55 | 1.24 to 1.94 | <0.001 |
| | Adjusted risk—model 2 | 1.31 | 1.04 to 1.65 | 0.002 | 1.76 | 1.40 to 2.21 | <0.001 | 1.31 | 1.11 to 1.54 | <0.001 | 1.36 | 1.09 to 1.71 | <0.001 |
| | Adjusted risk—model 3 | 1.35 | 1.07 to 1.71 | 0.001 | 1.83 | 1.46 to 2.30 | <0.001 | 1.31 | 1.12 to 1.54 | <0.001 | 1.37 | 1.09 to 1.73 | <0.001 |
| | Adjusted risk—model 4 | 1.19 | 0.93 to 1.50 | 0.066 | 1.40 | 1.10 to 1.77 | <0.001 | 1.17 | 0.99 to 1.38 | 0.015 | 1.10 | 0.87 to 1.40 | 0.274 |

Model 1 adjusts for maternal age, ethnicity, gestational diabetes and systolic blood pressure.
Model 2 is model 1 plus the other two socioeconomic status indicators (eg, the maternal unemployment column is adjusted for maternal education and partnership).
Model 3 is model 2 plus maternal body mass index as a potential mediator.
Model 4 is model 3 plus maternal smoking history as an additional mediator.
All models for the whole sample are adjusted for parity.

**Table 5** Risk of being born small for gestational age (birth weight <10th percentile for gestational age) by partner's employment and stratified by parity in the University Hospital Southampton maternity population-based cohort (singleton live births 2004–2016)

| Sample | Model | Unemployed partner at first antenatal appointment | | |
| --- | --- | --- | --- | --- |
| | | OR | 99% CI | P value |
| Whole sample, n mothers=41 841 | Unadjusted risk | 1.60 | 1.42 to 1.81 | <0.001 |
| | Adjusted risk—model 1 | 1.61 | 1.41 to 1.82 | <0.001 |
| | Adjusted risk—model 2 | 1.43 | 1.26 to 1.63 | <0.001 |
| | Adjusted risk—model 3 | 1.44 | 1.26 to 1.64 | <0.001 |
| | Adjusted risk—model 4 | 1.26 | 1.11 to 1.44 | <0.001 |
| Primiparous women only, n births=26 498 | Unadjusted risk | 1.76 | 1.50 to 2.06 | <0.001 |
| | Adjusted risk—model 1 | 1.60 | 1.36 to 1.88 | <0.001 |
| | Adjusted risk—model 2 | 1.44 | 1.21 to 1.70 | <0.001 |
| | Adjusted risk—model 3 | 1.45 | 1.22 to 1.71 | <0.001 |
| | Adjusted risk—model 4 | 1.33 | 1.12 to 1.58 | <0.001 |
| Multiparous women only, n births=15 343 | Unadjusted risk | 1.65 | 1.36 to 2.01 | <0.001 |
| | Adjusted risk—model 1 | 1.62 | 1.32 to 1.98 | <0.001 |
| | Adjusted risk—model 2 | 1.42 | 1.15 to 1.74 | <0.001 |
| | Adjusted risk—model 3 | 1.43 | 1.16 to 1.76 | <0.001 |
| | Adjusted risk—model 4 | 1.15 | 0.93 to 1.43 | 0.088 |

Model 1 adjusts for maternal age, ethnicity, gestational diabetes and systolic blood pressure.
Model 2 is model 1 plus maternal education and employment.
Model 3 is model 2 plus maternal body mass index as a potential mediator.
Model 4 is model 3 plus maternal smoking history as an additional mediator.
All models for the whole sample are adjusted for parity.

## Comparison with other studies

The evidence for SES inequalities by maternal educational attainment, employment and partner's employment in the risk of SGA is consistent with the literature, and the third analysis shows that these associations remain robust after limiting the sample to one record per mother. Within a systematic review of socioeconomic disparities in birth outcomes conducted in 2010,[7] 6 of the 9 (66%) studies of SGA and maternal education reported a significant association, in addition to single studies finding an association for maternal[41] and paternal employment.[42] Part of the complexity in the relationship between maternal SES and SGA results from many analyses using only one measure of SES, with maternal education[15 43] and employment[44] being the main indicators used. Factors related to the mother's partner are usually excluded, due to a lack of appropriate data or small sample sizes, despite the potential of these factors to influence pregnancy conditions and outcomes.[45] Whether the mother has a partner or not is largely overlooked as a risk factor in this area, with the exception of Kleijer et al,[46] who found that single mothers are at higher risk of SGA. The final estimates of SES inequalities in this study are adjusted for other SES indicators, suggesting that there are multiple pathways through which SES is linked to gestational growth.

Since the publication of Blumenshine et al's systematic review,[7] there has been an increased focus on how SES differences in weight outcomes at birth and during early life may be mediated through maternal BMI and smoking. In a Dutch cohort, maternal smoking and height during pregnancy were reported to explain 75% of the difference in risk of SGA between mothers with low and high education.[15] In an Australian cohort, maternal smoking and the BMI of both parents were reported to explain 83.5% of SES differences in their children's BMI Z-score at age 10–11 years.[47] In the present analysis, accounting for maternal smoking reduced the magnitude of the SGA risk difference by SES from a 36% increase in risk to 20% among mothers without a university degree, and from a 42% to 27% increase in risk among unemployed mothers. Maternal smoking also explained the relatively high risk of SGA among single mothers. This attenuation corroborates previous research indicating that single mothers are more likely to smoke, and that this may be related to the level of stress that they report, relative to partnered mothers.[48] Single mothers may be relying on smoking as a means of stress relief or management during pregnancy, and smoking cessation and support programmes may be effective in reducing inequalities in birth outcomes as a result.

To our knowledge, there has been no analysis of socioeconomic inequality time trends in SGA from the mid-2000s onwards in England. Inequalities in birth weight (adjusting for gestational age) were stable between

1961 and 2000 in a regional city-based study in North East England,[30] and the same is found between 2004 and 2016 in this study. The stability of SES inequalities in SGA implies that further interventions and initiatives are required to narrow SES inequalities in SGA births.

Our hypothesis was that the extent of SES inequalities in the risk of SGA may differ by parity, as the birth of the first child is a period which brings about significant physiological, lifestyle and social changes, in addition to post-partum weight retention.[18] An analysis of birth register data in Norway found that mothers who had several SGA births were characterised by low educational attainment and partners employed in low SES occupations.[19] In the present analysis, the strength of the association between SES indicators and the risk of SGA varied between primiparous and multiparous women, with education inequalities being greater for multiparous women, and employment inequalities being greater for primiparous women. The explanation may be that more advantaged women are economically able to leave the workforce after their first birth when planning further pregnancies (thus attenuating the differences between those in and outside employment when having subsequent births), while educational differences in terms of health behaviours, health literacy and mental well-being are risk factors of having repeat or new SGA outcomes.[49] This group may benefit from additional support following the birth of their first baby to promote mental and physical well-being, access appropriate services, enhance health literacy and facilitate healthy behaviours.

## Strengths and limitations of the study

This study benefits from a large regionally representative sample over many years. The exposure measures are prospectively collected in the course of routine antenatal care. As data from the local hospital system are used, there is no selection bias which may arise from participation in a research cohort, and the sample is therefore representative of all those receiving care under this NHS site. The outcome (SGA) is derived from birth weight, which is objectively measured by a health professional at birth. The most recent birth centiles for England and Wales were used[29] to reflect changes in birth weight since the 1990 birth centiles.[50] The measures of SES used are also collected within the usual course of NHS care before birth, so the results may be used to inform risk stratification interventions at or following the booking appointment to curtail SGA births and other associated adverse health outcomes. The antenatal booking appointment is a critical point for intervention as health professionals see all mothers receiving care under the NHS. The results herein find that women who report low educational qualifications are unemployed, or their partner is unemployed at this stage are at higher risk of SGA delivery. These groups, as well as women with no partners and/or other social support at the time of the booking appointment, may then be referred for additional support to minimise the risk of an SGA birth and other adverse maternal and health outcomes. A limitation of our dataset is that such processes were not electronically recorded and hence not included in our analyses. In addition, as this research is based on a cohort, we cannot infer that SES has a causal effect on SGA risk.

Some potential risk factors were not adjusted for in this study due to inconsistency of data for those specific variables as captured routinely in antenatal care, including diet during pregnancy and alcohol intake. These factors may also mediate the effect of SES on SGA risk, wherein disadvantaged SES groups could be more likely to engage in risky health behaviours. Other important SES factors such as sector of employment and income have been related to SGA outcomes in previous research,[9] but are also not routinely collected in antenatal practice. The same is true for other measures of deprivation level such as housing, transportation methods and access to healthcare and other facilities.

For the parity analysis, we did not account for the length of the interpregnancy interval which has been related to birth outcomes previously.[51 52] It was not possible to control for this in our study due to a lack of data on this variable in the whole sample, because we have included the first pregnancy in the database per mother and some multiparous mothers would have given birth before the study period, or at other hospitals; hence, this information is lacking for them. In addition, this analysis did not account for characteristics of the residential environments mothers lived in during pregnancy. Systematic reviews indicate that social, built and air characteristics of the environment experienced during pregnancy are strongly associated with birth outcomes,[6 53] and this will be addressed in a follow-up study on the associations between environmental characteristics and birth outcomes for the cohort.

As the data used in this study are limited to a hospital serving the city of Southampton and the surrounding region, the results may not apply to hospitals serving populations with differing characteristics. Southampton is a provincial urban city which is more deprived than the average Local Authority in England, although the surrounding area (Hampshire) is relatively affluent.[54] Southampton has a similar ethnicity profile to the rest of England and Wales,[55] but with a relatively large university student population, and women in Southampton are under-represented in managerial, administrative and professional occupations, relative to others in England.[56] As a result, findings from this study may not be replicated using healthcare records in areas with predominantly rural populations, or areas with non-student and managerial populations.

The UHS is a regional maternity centre to which high-risk pregnancies may be referred leading to potential over-representation of them. We have addressed this through excluding pregnancies booking in the UHS system after 24-week gestation. Mothers attending later than this date may have been referred to UHS due to their pregnancy being identified as high risk. We have

also conducted sensitivity analyses restricting the sample to those who were living in the city of Southampton at the time of birth, and there was no significant difference in effect sizes. The proportion of mothers in employment (64%) and with a university degree (28%) was similar in our cohort in comparison to Census figures for Southampton women aged 20–39 (69%) and 16–34 (29%), respectively, indicating that our sample is representative of the catchment area for the UHS.[57 58]

### Implications for research and practice

The persistence of educational and employment inequalities in the risk of SGA found within this study justifies further interventions and initiatives in order to narrow SES inequalities in the risk of SGA, and subsequently their long-term adverse health impact. The antenatal booking appointment offers an opportune moment for risk stratification and signposting of additional support for women with low educational qualification, in unemployed households and low social support. Smoking appeared as a potential mediator for SES inequalities in this study, despite support in smoking cessation being offered in the course of NHS care.[59] This suggests that further support is required for mothers of low SES, and preconception and interconception programmes may have the added benefit of reducing the extent of SES inequalities in SGA, in addition to overall SGA rates. For research, this study aligns with recent calls to incorporate paternal/partner influences in developmental health research,[45] in that similar levels of SGA risk are found for maternal and partner unemployment. Research in this area should adopt a more family-centred approach in relation to offspring health outcomes, taking into account contributing exposures from others within the household structure (partners and siblings).

## CONCLUSIONS

This study confirms that SES indicators, including educational attainment, employment status and single motherhood, are strongly and independently associated with the risk of SGA birth, and they are not narrowing over time. Maternal smoking appears to play a significant role in these inequalities, particularly for lone mothers. However, the associations between educational attainment and employment status with SGA risk remain strong even after accounting for maternal smoking and BMI. Inequalities in SGA risk by maternal educational attainment appear greater for multiparous compared with primiparous women, while the opposite is true by maternal and partner employment status. Further research is needed to identify critical windows of opportunity (preconception/pregnancy/interconception) and effective interventions in order to narrow these inequalities. Prevention programmes targeting socioeconomically disadvantaged women which incorporate smoking cessation and social support are vital to tackling health inequalities in SGA.

**Author affiliations**
¹School of Primary Care and Population Sciences, Faculty of Medicine, University of Southampton, Southampton, UK
²Geography & Environment, University of Southampton, Southampton, UK
³Public Health, Southampton City Council, Southampton, UK
⁴Department of Obstetrics and Gynaecology, University of Copenhagen, Roskilde, Denmark
⁵London Women's Clinic, London, UK
⁶School of Economic, Social and Political Science, Faculty of Social Sciences, University of Southampton, Southampton, UK
⁷Institute of Developmental Sciences, Academic Unit of Human Development and Health, Faculty of Medicine, University of Southampton, Southampton, UK
⁸NIHR Southampton Biomedical Research Centre, University of Southampton and University Hospital Southampton NHS Foundation Trust, Southampton, UK

**Acknowledgements** The authors thank Ravita Taheem (Senior Public Health Practitioner, Southampton City Council) for her input in the conception of this paper, David Cable (Electronic Patient Records Implementation and Service Manager) and Florina Borca (Senior Information Analyst for R&D, NIHR Southampton Biomedical Research Centre) at University Hospital Southampton NHS Foundation Trust for their support in extracting the data used in this study.

**Contributors** NAA is the Principal Investigator of the project, and acts as the guarantor of this study. SW, NZ, PR, DS, DC, NM, MH and NAA contributed to study conception and design. NM provided input on the statistical analysis for this study. SW conducted the statistical analyses, and drafted the initial report. NZ checked the accuracy of the reported estimates from the statistical models. All authors contributed to the interpretation of data and revising the manuscript critically for important intellectual content. All authors read and approved the final manuscript.

**Funding** This research is supported by an Academy of Medical Sciences and Wellcome Trust grant to NAA [Grant no: HOP001\1060], and the National Institute for Health Research through the NIHR Southampton Biomedical Research Centre. The research funders had no input on research design or manuscript drafting. MAH is supported by the British Heart Foundation and the National Institute for Health Research through the NIHR Southampton Biomedical Research Centre.

**Competing interests** NAA had financial support from the Academy of Medical Sciences/Wellcome Trust and the NIHR Southampton Biomedical Research Centre for the submitted work; no financial relationships with any organisations that might have an interest in the submitted work in the previous three years; NAA is a member of the National Institute for Health and Care Excellence Antenatal Care Guideline Committee; no other relationships or activities that could appear to have influenced the submitted work.

**Patient consent for publication** Not required.

**Ethics approval** This study used anonymised antenatal record data supplied by University Hospital Southampton Trust. This analysis forms part of a research project reviewed and approved by the University of Southampton Faculty of Medicine Ethics Committee (ref 24433) and the National Health Service Health Research Authority (ref 242031).

**Provenance and peer review** Not commissioned; externally peer reviewed.

**Data sharing statement** The authors' ethical approval from the Faculty of Medicine Ethics Committee, University of Southampton (Reference number 24433) restricts public sharing of the data used in this study. The data owner is University Hospital Southampton NHS Trust. Contact NAA to request data access beyond that included in the manuscript. Further ethical and research governance approval may be required.

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
