## [Reviewer comments · BMJ Open]

ARTICLE DETAILS

TITLE (PROVISIONAL)	Are socioeconomic inequalities in the incidence of small-for-gestational-age birth narrowing? Findings from a population-based cohort in the South of England
AUTHORS	Wilding, Sam; Ziauddeen, Nida; Roderick, Paul; Smith, Dianna; Chase, Debbie; Macklon, Nick; McGrath, Nuala; Hanson, Mark; Alwan, Nisreen

VERSION 1 – REVIEW

REVIEWER	L Smith University of Leicester
REVIEW RETURNED	12-Nov-2018

GENERAL COMMENTS	This is a really interesting paper addressing a very important issue and with socioeconomic data often not available on a large scale. The data are extremely detailed for SES measures and other factors which is rarely seen in any routine datasets. The statistical approach is appropriate and takes into account the complexity of the dataset with complex analyses. However I am concerned that because it is not a geographically based population with no information on changes in the cohort over the 14 year period there are some strong concerns about the findings and whether they are generalizable to another population. Some sensitivity analyses could address these issues. I have outlined this and some small other suggestions below. Abstract Participants says 23+ but actually 24+ in the analyses – this should refer to the data in the main models Introduction The authors refer to “Public health policy aims to narrow SES inequalities in birth outcomes over time, and there is reason to believe that inequalities in SGA may have changed since the early 2000s” however there is no reference to any policies in place or evidence of inequalities in SGA. This would greatly benefit from referencing. Methods and Results My main worry is that this is a hospital-based cohort of referrals. The hospital under study has a neonatal intensive care unit with provision of neonatal surgery and consequently will have referrals from a wide area for these services. This practice may have changed over time and with no underlying data on the population over the time period we cannot assess whether the lack of change in the inequalities in SGA is due to actual lack of change in the SES inequalities or to a change in the referral of patients over the time period. A sensitivity analyses limiting the analyses as far as
--

	possible to a geographical area where it is known that a high rate of the population are cared for by that hospital (e.g. geographical small areas included where it is known that 99% of women are referred to that hospital for care) would help the reliability of the findings. It also means the findings are more appropriate to a higher than average risk group as the hospital takes high risk women and low risk women may be cared for elsewhere. Some information on the trends in the SES factors over time in the population under study would help in some way towards understanding changes in the population of women being cared for although still would be difficult to distinguish between trends in these factors at a national level and changes in trends in the population provided for. I'm unsure why variables such as education are dichotomised into degree/no degree – the dose response rate in this variable would be of great interest and may suggest an alternative cut-off. Without the data on this it is hard to understand. Regarding employment status, how are mothers looking after the home classified. This may be different for mothers who were previously unemployed or previously in work. For SGA why was the tenth centile chosen. There are many definitions and a sensitivity analyses of alternative centile cut-offs may be of interest to understand where the inequalities are at their greatest. The BMI variable was treated as a continuous variable and yet may be very skewed. I would suggest treating it as a categorical variable to assess the relationship with SES. Overall I think there are some sensitivity analyses that could improve the reliability of the findings and reassure the reader that they are generalizable on a wider scale. Discussion Overall the discussion is appropriate but more detailed discussion of the limitations of the population is needed.
--	--

REVIEWER	Ana Daniela I. de Sadovsky Universidade Federal do Espirito Santo - Vitoria - ES. Brazil
REVIEW RETURNED	02-Jan-2019

GENERAL COMMENTS	The manuscript comprises an original work with clear focus on socioeconomic status and SGA. The author has taken care of ethical considerations while developing the manuscript. The writing style and language are impressive. The references are judiciously used in the manuscript. Some topics not explained in the methodology were duly clarified during the presentation of the results, therefore there is no need for modification. The analysis of the data and interpretation of the results is good. Results clearly demonstrated and of great importance for the scientific community, mainly related to the inadequate birth outcomes in susceptible and socioeconomically underprivileged populations. However, I think that it requires a better discussion through the presented results, with adequate sampling and well-done statistics. The main focus was education and maternal or family employment, reflecting the socio-economic status of SGA outcome. There is a great meta-analysis article with data
--

	(including from England) that relates SGA to maternal schooling that wasn't quoted and I feel like it would be interesting to have a look. Follows: Ruiz M et al. Mother's education and the risk of preterm and small for gestational age birth: the DRIVERS meta-analysis of 12 European cohorts. J Epidemiol Community Health. 2015 Sep; 69 (9): 826-33. In addition, when related to parity, we have literature with important questions about the association of multiparity: the number of children or the interval between the gestations, a fact that was not addressed during the discussion. Suggestion: Kozuki N et al. The associations of birth intervals with small-for-gestational-age, preterm, and neonatal and infant mortality: the meta-analysis. BMC Public Health. 2013; 13 Suppl 3: S3. doi: 10.1186 / 1471-2458-13-S3-S3. Epub 2013 Sep 17. There was no discussion or even mention of the limitations regarding the quality of maternal employment or even income, what I feel should be included especially when we have several articles associating low weight births with heavy work and with income as a important economic factor. The influence on income quintiles changes has already been shown to be related to the significant reduction in the prevalence of SGA in bith cohorts in Brazil. Sadovsky et al. LBW and IUGR temporal trend in 4 population-based birth cohorts: the role of economic inequality. BMC Pediatrics (2016) 16:115. The introduction can be shortened and the discussion expanded on the most important topics that have been found.
--	--

REVIEWER	Anura W.G. Ratnasiri Policy and Benefits Branch California Department of Health Care Services 1501 Capitol Ave, Suite. 71.4104, MS 4600 P.O. Box 997417 Sacramento, CA 95899-7417
REVIEW RETURNED	01-Apr-2019

GENERAL COMMENTS	Thank you for the opportunity to review this manuscript. The study topic was "Are socioeconomic inequalities in the incidence of small-for-gestational-age birth narrowing? Findings from a population-based cohort in the South of England". The data was from the Population based cohort study used 65,825 singleton live births born to mothers aged ≥18 years between 23 and 42 weeks gestation born at University Hospital Southampton, UK, between 2004 and 2016. Authors examined differences in SGA risk by SES indicators. SGA is an extremely important public health topic, especially given a reported significance in the past couple of years. The writing quality of this manuscript is thoughtful and it has good intentions. Tropic: Consistency of using small-for-gestational-age Abstract: PI spell out SES Introduction: Line 7: specific about maternal underweight and higher maternal obesity classes
--

	Attribute to Bakers hypothesis to emphasize the significance of SGA Methods: Include information on how much was smoked per day if available. If available, was a dose-response association noted? Do you have enough cohorts in cells when you run Model 4? Discussion: A major limitation is the lack of information about key maternal medical and obstetric conditions that predispose SGA or are other risk factors for SGA. If no maternal morbidity data were available, then it is important to highlight this in your limitations section. As a consequence, it is unclear whether the extent of any differences across age groups, education, employment status, ethnicities, and parity etc is (partly) explained by differences in maternal medical/obstetric characteristics which may differentially impact these subgroups. You may want to downplay your main take-home message and indicate that further work is needed to determine whether the differences across these subgroups is still present after accounting for these residual confounders/mediators. Please describe the SES indicators that were unmeasured, e.g. housing, transportation, access to health care. Consider to mention: Often, the 10th percentile on a fetal growth standard is used as a threshold for identifying a 'small' fetus. However, this has been criticized because it includes many infants who are constitutionally small and not at risk for poor perinatal outcome.
--	---

VERSION 1 – AUTHOR RESPONSE

Reviewer 1:

This is a really interesting paper addressing a very important issue and with socioeconomic data often not available on a large scale. The data are extremely detailed for SES measures and other factors which is rarely seen in any routine datasets. The statistical approach is appropriate and takes into account the complexity of the dataset with complex analyses. However I am concerned that because it is not a geographically based population with no information on changes in the cohort over the 14 year period there are some strong concerns about the findings and whether they are generalizable to another population. Some sensitivity analyses could address these issues. I have outlined this and some small other suggestions below.

Thank you for your positive comments. We attempt to address your concerns point by point below.

- [In the abstract] participants says 23+ but actually 24+ in the analyses – this should refer to the data in the main models

Thank you. 23 has been corrected to 24 in the abstract

- [In the introduction] The authors refer to “Public health policy aims to narrow SES inequalities in birth outcomes over time, and there is reason to believe that inequalities in SGA may have changed since the early 2000s” however there is no reference to any policies in place or evidence of inequalities in SGA. This would greatly benefit from referencing.

Thank you. This sentence has been changed to “In England, public health policy aims to narrow SES inequalities in birth outcomes over time [22,23], and changes in the extent of inequalities in SGA have been noted in other European countries since the early-2000s [24]” - lines 96-98

The Local Government Association’s ‘fit for and during pregnancy’ policy document [22] has been included as evidence of government policy aiming to reduce inequalities in birth outcomes. Also, the Public Health Outcomes Framework for England’s second high level outcome is to reduce differences in life expectancy and healthy life expectancy between communities [23]. One of the indicators to measure this outcome is ‘low birth weight of term babies’. Both have now been cited in the introduction. Kana et al’s paper on the “Impact of the global financial crisis on low birth weight in Portugal: a time-trend analysis” [24] has been included as evidence of changes in the extent of socioeconomic inequalities in birth outcomes in Portugal in a time frame (1995-2014) that overlaps this study (2004-2016).

- My main worry is that this is a hospital-based cohort of referrals. The hospital under study has a neonatal intensive care unit with provision of neonatal surgery and consequently will have referrals from a wide area for these services. This practice may have changed over time and with no underlying data on the population over the time period we cannot assess whether the lack of change in the inequalities in SGA is due to actual lack of change in the SES inequalities or to a change in the referral of patients over the time period. A sensitivity analyses limiting the analyses as far as possible to a geographical area where it is known that a high rate of the population are cared for by that hospital (e.g. geographical small areas included where it is known that 99% of women are referred to that hospital for care) would help the reliability of the findings. It also means the findings are more appropriate to a higher than average risk group as the hospital takes high risk women and low risk women may be cared for elsewhere. Some information on the trends in the SES factors over time in the population under study would help in some way towards understanding changes in the population of women being cared for although still would be difficult to distinguish between trends in these factors at a national level and changes in trends in the population provided for.

Thank you for raising this important point. We appreciate and understand the concern about the representativeness of the UHS maternity database population, given that neonatal intensive care is also provided at the hospital. The concern is that we may have a higher than average proportion of high-risk pregnancies captured within our sample, and that this will affect the extent to which our findings can be applied to other areas. It bears noting that the majority of pregnancies cared for at UHS receive routine antenatal care in the community, however the maternity system used to record information is the UHS system and most women end up having their delivery there. As mentioned in the manuscript on lines 118-120, we have excluded late bookings (after 24 weeks gestation) from the analysis. Discussions with the midwifery team at UHS have indicated that late referrals may be an indication that women are referred later in their pregnancy to this regional centre for high-risk pregnancy or complications.

Since submitting the manuscript, we have successfully linked birth records to place of residence at birth for children who are cared for by Solent Health’s Health Visitors. From a sample of 65,412 births, this amounts to 32,147 births (49%) which were successfully linked, with the remainder likely belonging to women resident in Hampshire (the surrounding county). Limiting these records to children who had their first health visitor appointment at an address within Southampton results in 30,663 births. In this reduced sample, the adjusted odds ratios are similar to that in the full sample, suggesting that we do not have an over-representation of high-risk pregnancies in this analysis. For example, the adjusted odds ratios for maternal college and school education was 1.10 (99% CI 1.00-1.22) and 1.30 (1.17-1.45) in the whole sample respectively, compared to 1.10 (99% CI 0.95-1.27)

and 1.29 (95% CI 1.11-1.51) in the Southampton sample. The intervals for the odds ratios are more disparate for maternal unemployment (99% CI 1.16-1.38 vs 1.23-1.56) and partner's employment (99% CI 1.16-1.38 vs 1.23-1.56), but the intervals overlap, and lone motherhood is not associated with SGA risk in either sample (99% CI 0.93-1.20 vs 0.86-1.20). This supplementary analysis is explained in lines 260-272, and included in Supplementary Table 3.

Based on England's 2011 Census figures for Southampton, we do not believe there is strong evidence that our sample is unrepresentative of women of reproductive age in Southampton. Comparing our sample to tables from the 2011 Census, 64% of our cohort are in employment compared to 69% of women aged 20-39 in Southampton (<https://www.nomisweb.co.uk/census/2011/dc6107ew>). Comparatively, 29% of our sample reported having a university degree, compared to 28% of women aged 16-34 in Southampton (<https://www.nomisweb.co.uk/census/2011/dc5102ew>). This information has been included in the discussion in lines 432-435.

We have included trends in SES factors by year in the cohort in a supplementary figure 1. Overall, the proportion reporting having a university degree increased (from 21% in 2004 to 35% in 2016), maternal employment increased (from 66% to 71%), partner's employment was stable (92% in both 2004 and 2016), and lone motherhood at the start of pregnancy decreased (9% to 6%). Given that the model adjusts for cohort year, a change in the proportion of SES groups will not affect the estimates of SES inequality.

- I'm unsure why variables such as education are dichotomised into degree/no degree – the dose response rate in this variable would be of great interest and may suggest an alternative cut-off. Without the data on this it is hard to understand.

Thank you. We have now broken down the non-degree holding sample into those with a college qualification (A levels), and those with a secondary school (GCSE) or lower qualification. This is now stated in the methods section under "Assessment of SES exposures" – lines 132-134. We did not have sufficient numbers to look at more refined categories of educational qualifications. We have redone the analysis using these new categories instead of the previous two.

- Regarding employment status, how are mothers looking after the home classified. This may be different for mothers who were previously unemployed or previously in work.

Thank you. Mothers looking after the home will be classified as unemployed, because this is how the raw variable is measured. Therefore, it is not possible to identify home-keepers specifically. Information on the raw variable can be found in lines 134-136.

- For SGA why was the tenth centile chosen. There are many definitions and a sensitivity analyses of alternative centile cut-offs may be of interest to understand where the inequalities are at their greatest.

Thank you. We have chosen to use this cut-off as it is the definition used by national guidelines and surveillance in England (please see references below), as we wanted to ensure our findings are relevant to policy and practice:

Royal College of Obstetricians and Gynaecologists. Small-for-Gestational-Age Fetus, Investigation and Management (2013)

<https://www.rcog.org.uk/en/guidelines-research-services/guidelines/gtg31/>
National Maternity and Perinatal Audit: Clinical Report 2017:

[http://www.maternityaudit.org.uk/downloads/RCOG%20NMPA%20Clinical%20Report\(web\).pdf](http://www.maternityaudit.org.uk/downloads/RCOG%20NMPA%20Clinical%20Report(web).pdf)

We have also included a reference to Schlaudecker et al (Small for gestational age: Case definition & guidelines for data collection, analysis, and presentation of maternal immunisation safety data), which identifies the 10% definition as most common in the literature, thus allowing for comparisons across the research evidence.

We have now included the phrase “In line with World Health Organisation guidelines, UK guidelines and common practice, SGA is defined as a birth weight lower than the 10th percentile compared to others born at the same number of weeks gestation in the sex-specific reference centiles” in the methods section with the appropriate references – lines 151-154

- The BMI variable was treated as a continuous variable and yet may be very skewed. I would suggest treating it as a categorical variable to assess the relationship with SES.

Thank you. BMI is now treated as a categorical variable (<18.5 underweight; 18.5-24.9 normal weight; 25-29.9 overweight; 30+ obese) in Tables 1-5 and Figure 1. Changing the way BMI was measured in the analysis did not change the extent of SES inequalities.

We have changed this in the methods section under “Assessment of confounder and mediator variables” which now reads:

“Maternal BMI was categorised as underweight (<18.5), normal (18.5-24.9), overweight/pre-obese (25.0-29.9) and obese (30+) [26], and treated as a categorical variable in all analyses.”-lines 166-168

- Overall I think there are some sensitivity analyses that could improve the reliability of the findings and reassure the reader that they are generalizable on a wider scale.

Thank you. We have conducted sensitivity analysis around the issue of representativeness as mentioned above. The results of this analysis are stated in lines 260-272 and presented in Supplementary Table 3.

- Overall the discussion is appropriate but more detailed discussion of the limitations of the population is needed.

We have included the following statement in the discussion:

The UHS is a regional maternity centre to which high-risk pregnancies may be referred leading to potential over-representation of them. We have addressed this through excluding pregnancies booking in the UHS system after 24 weeks gestation. Mothers attending later than this date may have been referred to UHS due to their pregnancy being identified as high-risk. We have also conducted sensitivity analyses restricting the sample to those who were living in the city of Southampton at the time of birth, and there was no significant difference in effect sizes. The proportion of mothers in employment (64%) and with a university degree (28%) were similar in our cohort in comparison to Census figures (which year) for Southampton women aged 20-39 (69%) and 16-34 (29%), respectively, indicating that our sample is representative of the catchment area for the UHS [47,48]. – lines 427-435.

Reviewer 2:

The manuscript comprises an original work with clear focus on socioeconomic status and SGA. The author has taken care of ethical considerations while developing the manuscript. The writing style and language are impressive. The references are judiciously used in the manuscript.

Some topics not explained in the methodology were duly clarified during the presentation of the results, therefore there is no need for modification.

The analysis of the data and interpretation of the results is good. Results clearly demonstrated and

of great importance for the scientific community, mainly related to the inadequate birth outcomes in susceptible and socioeconomically underprivileged populations.

However, I think that it requires a better discussion through the presented results, with adequate sampling and well-done statistics.

Thank you for your positive comments. We attempt to address your concerns point by point below.

- The main focus was education and maternal or family employment, reflecting the socioeconomic status of SGA outcome. There is a great meta-analysis article with data (including from England) that relates SGA to maternal schooling that wasn't quoted and I feel like it would be interesting to have a look. Follows: Ruiz M et al. Mother's education and the risk of preterm and small for gestational age birth: the DRIVERS meta-analysis of 12 European cohorts. *J Epidemiol Community Health*. 2015 Sep; 69 (9): 826-33.

Many thanks for this helpful suggestion. We have inserted the following sentence in the introduction, which references this paper and the LBW and IUGR temporal trend paper mentioned below:

“Disadvantaged SES groups (in terms of education and income) typically experience greater rates of SGA births [8,9]” – lines 75-76.

- In addition, when related to parity, we have literature with important questions about the association of multiparity: the number of children or the interval between the gestations, a fact that was not addressed during the discussion. Suggestion: Kozuki N et al. The associations of birth intervals with small-for-gestational-age, preterm, and neonatal and infant mortality: the meta-analysis. *BMC Public Health*. 2013; 13 Suppl 3: S3. doi: 10.1186 / 1471-2458-13-S3-S3. Epub 2013 Sep 17.

Thank you. We acknowledge the importance of the inter-pregnancy interval as a potential confounder of the association between SES and SGA. Unfortunately we only have information on the inter-pregnancy interval for women who have at least two of their pregnancies recorded in the dataset, therefore we cannot take account of this variable in the whole sample of live singleton pregnancies, or in the parity stratified analysis included in this paper as we have included the first recorded birth per mother in the dataset.

However, a separate analysis of a subset of this cohort of women with two recorded pregnancies in the dataset showed an association of an inter-pregnancy interval of 12-23 months with lower risk of SGA at the second pregnancy compared with an interval of 24-35 months in mothers whose first and second pregnancies were recorded. Please see the published conference abstract cited below:

Ziauddeen N, Roderick PJ, Macklon NS, Alwan NA. LB5 Is the duration of the preceding interpregnancy interval associated with offspring's size at birth?—analysis of a UK population-based cohort. *Journal of Epidemiology & Community Health*. 2018 Sep 1;72(Suppl_1).

In light of this, we have included the following sentences in the discussion, referencing the recommended meta-analysis and our published abstract on this:

“For the parity analysis, we did not account for the length of the interpregnancy interval which has been related to birth outcomes previously [50,51]. It was not possible to control for this in our study due to a lack of data on this variable in the whole sample, because we have included the first pregnancy in the database per mother and some multiparous mothers would have given birth before the study period, or at other hospitals, hence this information is lacking for them”

- There was no discussion or even mention of the limitations regarding the quality of maternal employment or even income, what I feel should be included especially when we have several articles associating low weight births with heavy work and with income as a important economic factor. The influence on income quintiles changes has already been shown to be related to the significant reduction in the prevalence of SGA in bith cohorts in Brazil. Sadovsky et al. LBW and IUGR temporal trend in 4 population-based birth cohorts: the role of economic inequality. BMC Pediatrics (2016) 16:115.

Thank you. We have included the following sentences in the discussion, which references this paper: “Other important SES factors such as sector of employment and income have been related to SGA outcomes in previous research [9], but are also not routinely collected in antenatal practice. The same is for other measures of deprivation level such as housing, transportation methods and access to healthcare and other facilities” – Lines 403-407.

- The introduction can be shortened and the discussion expanded on the most important topics that have been found.

Thank you. We have now shortened the introduction to 1 side of A4, and expanded the discussion of limitations responding to the reviewers’ comments.

Reviewer 3:

Thank you for the opportunity to review this manuscript. The study tropic was “Are socioeconomic inequalities in the incidence of small-for gestational-age birth narrowing? Findings from a population-based cohort in the South of England”. The data was from the Population based cohort study used 65,825 singleton live births born to mothers aged ≥ 18 years between 23 and 42 weeks gestation born at University Hospital Southampton, UK, between 2004 and 2016. Authors examined differences in SGA risk by SES indicators.

SGA is an extremely important public health topic, especially given a reported significance in the past couple of years. The writing quality of this manuscript is thoughtful and it has good intentions.

Thank you for your positive comments. We attempt to address your concerns point by point below.

- [In the abstract] spell out SES

Thank you. Changed SES to “socioeconomic”

- Line 7: specific about maternal underweight and higher maternal obesity classes Attribute to Bakers hypothesis to emphasize the significance of SGA

Thank you. We have adjusted for maternal body mass index in all our models, and have referred to the Developmental Origins of Health and Disease (the field arising from David Barker’s work) in line 68.

- Include information on how much was smoked per day if available. If available, was a doseresponse association noted?

This information has now been included in Table 1. Smoking was reported as either <10 cigarettes per day, 10-20 per day and 20+ per day. Proportions of SGA births were as follows: 16% for smoking <10 cigarettes per day, 18% for smoking 10-20 cigarettes per day, and 22% for smoking >20 cigarettes per day. Hence, a dose-response relationship is apparent, however this is based on selfreporting

so there is a possibility of under-reporting.

- Do you have enough cohorts in cells when you run Model 4?
There were no problems with model convergence.

- A major limitation is the lack of information about key maternal medical and obstetric conditions that predispose SGA or are other risk factors for SGA. If no maternal morbidity data were available, then it is important to highlight this in your limitations section. As a consequence, it is unclear whether the extent of any differences across age groups, education, employment status, ethnicities, and parity etc is (partly) explained by differences in maternal medical/obstetric characteristics which may differentially impact these subgroups. You may want to downplay your main take-home message and indicate that further work is needed to determine whether the differences across these subgroups is still present after accounting for these residual confounders/mediators.

Thank you. The model estimates account for mother's systolic blood pressure at booking. In addition, we have now accounted for gestational diabetes and gestational hypertension in the models. The inclusion of these health indicators has had no significant effect on the SES inequality estimates.

- Please describe the SES indicators that were unmeasured, e.g. housing, transportation, access to health care.

In line with the comment along the same lines from reviewer 2, we have included this section in the discussion:

“Other important SES factors such as sector of employment and income have been related to SGA outcomes in previous research [9], but are also not routinely collected in antenatal practice. The same is true for other measures of deprivation level such as housing, transportation methods and access to healthcare and other facilities.” – lines 403-407

- Consider to mention: Often, the 10th percentile on a fetal growth standard is used as a threshold for identifying a 'small' fetus. However, this has been criticized because it includes many infants who are constitutionally small and not at risk for poor perinatal outcome.

Thank you. A comment along the same lines was made by reviewer 1. Here is our response:

We have chosen to use this cut-off as it is the definition used by national guidelines and surveillance in England (please see references below), as we wanted to ensure our findings are relevant to policy and practice:

Royal College of Obstetricians and Gynaecologists. Small-for-Gestational-Age Fetus, Investigation and Management (2013)

<https://www.rcog.org.uk/en/guidelines-research-services/guidelines/gtg31/>
National Maternity and Perinatal Audit: Clinical Report 2017:

[http://www.maternityaudit.org.uk/downloads/RCOG%20NMPA%20Clinical%20Report\(web\).pdf](http://www.maternityaudit.org.uk/downloads/RCOG%20NMPA%20Clinical%20Report(web).pdf)

We have also included a reference to Schlaudecker et al (Small for gestational age: Case definition & guidelines for data collection, analysis, and presentation of maternal immunisation safety data), which identifies the 10% definition as most common in the literature, thus allowing for comparisons across the research evidence.

We have now included the phrase “In line with World Health Organisation guidelines, UK guidelines and common practice, SGA is defined as a birth weight lower than the 10th percentile compared to others born at the same number of weeks gestation in the sex-specific reference centiles” in the methods section with the appropriate references – lines 151-154